# Understanding Human Limits in Pattern Recognition: A Computational Model of Sequential Reasoning in Rock, Paper, Scissors

Logan Cross[1]*, Erik Brockbank[2]*, Tobias Gerstenberg[2], Judith E. Fan[2], Daniel L. K. Yamins[2], Nick Haber[3]
[1]Stanford Computer Science
[2]Stanford Department of Psychology
[3]Stanford Graduate School of Education

## Abstract

**How do we predict others from patterns in their behavior and what are the computational constraints that limit this ability? We investigate these questions by modeling human behavior over repeated games of rock, paper, scissors from Brockbank & Vul (2024). Against algorithmic opponents that varied in strategic sophistication, people readily exploit simple transition patterns (e.g., consistently playing rock after paper) but struggle to detect more complex sequential dependencies. To understand the cognitive mechanisms underlying these abilities and their limitations, we deploy Hypothetical Minds (HM), a large language model-based agent that generates and tests hypotheses about opponent strategies, as a cognitive model of this behavior (Cross et al., 2024). We show that when applied to the same experimental conditions, HM closely mirrors human performance patterns, succeeding and failing in similar ways. To better understand the source of HM's failures and whether people might face similar cognitive bottlenecks in this context, we performed a series of ablations and augmentations targeting different components of the system. When provided with natural language descriptions of the opponents' strategies, HM successfully exploited 6/7 bot opponents with win rates >80% suggesting that accurate hypothesis generation is the primary cognitive bottleneck in this task. Further, by systematically manipulating the model's hypotheses through pedagogically-inspired interventions, we find that the model substantially updates its causal understanding of opponent behavior, revealing how model-based analyses can produce testable hypotheses about human cognition.**

**Keywords:** theory of mind; pattern learning; social reasoning; large language models; rock, paper, scissors

## Introduction

The ability to predict others' behavior is central to social interaction (FeldmanHall & Shenhav, 2019; FeldmanHall & Nassar, 2021; Tamir & Thornton, 2018). To make these predictions, humans deploy sophisticated cognitive processes, called *Theory of Mind* (ToM), that infer the hidden causes underlying observed actions (Ho et al., 2022; Baker et al., 2017). In both collaborative and competitive interactions, humans make a broad suite of predictive inferences about the hidden mental states of other intentional agents, such as their motives (van Baar et al., 2022; Ullman et al., 2009; Wu et

al., 2023), strategies (Kleiman-Weiner et al., 2016), competence (Y. Xiang et al., 2023), and emotions (Ong et al., 2019). This socially predictive machinery also learns the statistical regularities of other people's behavior, such as the transition probabilities between common activities or emotional states (Thornton & Tamir, 2017, 2021).

How do we learn a predictive model of others from structured patterns in their past behavior? In controlled settings, researchers have investigated this question with iterated economic games (Allen et al., 2024; Camerer, 2011). In particular, *rock, paper, scissors* (RPS) has emerged as a "model system" for studying how people detect sequential patterns in others' behavior (Guennouni & Speekenbrink, 2022; Brockbank & Vul, 2021, 2024; Zhang et al., 2021). First, the game requires minimal expertise—instead, a player's success is a function of their ability to identify exploitable patterns in an opponent's play while avoiding such patterns in their own moves. Since the Nash Equilibrium is random behavior (Morgenstern & Neumann, 1953; Nash, 1950), the only way to perform better than chance is to exploit non-random dependencies in opponent's moves. Thus, a player's outcomes directly reflect their ability to acquire a strong predictive model of their opponent. Also, the sequential dependencies that a player might exploit in their opponent—can be precisely characterized (Dyson, 2019; Brockbank & Vul, 2021), allowing for fine-grained description of the strategic inferences a player makes about their opponent.

Research exploring the ability to predict and adapt to an opponent in the RPS game often pits human participants against algorithmic opponents that systematically vary in sequential dependencies dictating their actions (Dyson, 2019; Brockbank & Vul, 2021). This work shows that people exploit simple sequential patterns but exhibit limitations in adapting to more complex dependencies (Brockbank & Vul, 2024; Guennouni & Speekenbrink, 2022; Zhang et al., 2021; Forder & Dyson, 2016), consistent with domain-general challenges of adaptive adversarial reasoning (Guennouni & Speekenbrink, 2022; Batzilis et al., 2019; Stöttinger et al., 2014). Several mechanisms could explain these limitations. For instance, people may fail to consider unintuitive patterns ("hypothesis *generation*"). Alternatively, people may be unable to accurately confirm patterns in their opponent's moves due to the difficulty of reasoning in this context ("hypothesis *evaluation*"). Or, it may be that people struggle to *plan* the right move given knowledge of their opponent's strategy.

To better disentangle the role that these distinct processes play in identifying and exploiting structured opponent behav-

ior, we deploy the large language model (LLM) agent Hypothetical Minds (HM) as a cognitive model of strategic decision making in this task (Cross et al., 2024). We model human RPS data from Brockbank & Vul (2024), in which people failed to adapt to complex sequential patterns in opponent behavior. This approach allows us to test a broad range of hypotheses about the bottlenecks that impact opponent prediction in this setting. HM addresses the challenge of inferring other agents intentions' and adapting to them, and the components it relies on to do this are domain-general and do not require a pre-specified encoding of the problem space (other than the natural language explanation of the task), allowing it to be applied to a diverse set of collaborative, adversarial, and mixed-motive domains (Cross et al., 2024). In the RPS setting, HM generates and evaluates different hypotheses about the opponent's strategy in natural language, allowing the agent to act adaptively given the best explanation of the opponent's moves. Given that the space of possible opponent strategies is vast and unbounded, natural language provides a useful, domain-general parameterization for modeling human reasoning in this extensive hypothesis space.

In the first part of the paper, we show that the HM model is able to capture the pattern of selective adaption to simple but not complex RPS opponents exhibited by humans in Brockbank & Vul (2024). Critically, we show that this correspondence relies on the model's Theory of Mind component (i.e., it does not emerge from LLMs alone), suggesting that ToM inference about an opponent is key to capturing patterns of human behavior and that the model's approach to doing so offers a reasonable approximation of similar processes in humans. Next, we use the correspondence between model and human behavior to test a number of different manipulations that might impact performance in the current task. We show that the model's success in the task is fundamentally constrained by the *hypothesis generation* processes; given an accurate description of the opponent's strategy or the opportunity to identify the true strategy from a larger set, it performs close to ceiling against all but the most complex opponent that is difficult to predict from language based reasoning alone. Next, we demonstrate that the hypothesis generation bottleneck is not resolved by merely taking more samples or sampling more widely from the existing distribution; rather, enabling the model to succeed requires altering the model's distribution over possible opponent strategies to consider entirely new hypotheses. In this way, "teaching" the model to think about the problem in different ways enables it to learn patterns in opponent behavior that it previously failed to recognize.

## Methods

### Experimental Data

We analyzed data from Brockbank & Vul (2024), in which participants played rock, paper, scissors against algorithmic opponents exhibiting different sequential patterns. In each round, participants selected one of three moves (rock, paper, or scissors) and received points based on the game's standard payoff structure: winning yielded 3 points, ties yielded 0 points, and losses yielded $-1$ points. Each participant (N=218) played 300 rounds against one of seven bot opponents. After 300 rounds, participants were asked to describe their opponent's strategy.

The bot opponents chose moves according to different sequential dependencies with 90% probability. These sequential dependencies increased in complexity based on the number of previous events required to predict the bot's move.

**Previous move dependencies**: Four bots based their moves on either their own or their opponent's previous move. We define three types of transitions between consecutive RPS moves A and B as illustrated in Figure 1: (1) a positive transition (+) occurs when move B beats move A (rock→paper, paper→scissors, scissors→rock), (2) a negative transition (-) occurs when move B loses to move A (rock→scissors, paper→rock, scissors→paper), and (3) a stay transition (0) occurs when move B is identical to move A. These transition types can be applied to both an agent's own moves (self-transitions) or in relation to an opponent's previous move (opponent-transitions), forming the basis for the sequential dependencies in our algorithmic opponents:

- Self-transition($+$): Moves that would beat their previous move
- Self-transition($-$): Moves that would lose to their previous move
- Opponent-transition($+$): Moves that would beat the opponent's previous move
- Opponent-transition($0$): Repeating the opponent's previous move

**Previous outcome dependencies**: Two bots favored different transitions after a win, loss, or tie:

- Previous outcome($W0L+T-$): Stay transitions after wins, positive transitions after losses, negative after ties
- Previous outcome($W+L-T0$): Positive transitions after wins, negative after losses, stay after ties

**Previous outcome & transition dependencies**: One bot (Previous outcome, previous transition) chose moves based on both the previous outcome and its previous transition.

### Hypothetical Minds Architecture

We adapted the Hypothetical Minds architecture from Cross et al. (2024). The model consists of three primary components: Memory, Theory of Mind Module, and Decision Reflection (Figure 1). The **Memory** component maintains a log of agent plays, opponent plays, and rewards in each previous round. This information serves as input to both the Theory of Mind module and the Decision Reflection component.

The Theory of Mind (ToM) module implements a natural language approximation of Bayesian inference over latent variables to model other agents' behavior in a conceptually analogous way as Bayesian inverse planning (Baker et al., 2009). In multi-agent environments such as RPS, other agents' actions are influenced by latent variables $\Theta = \theta_1, \ldots, \theta_m$ (strategies, competence levels, etc.) that must be inferred from partial observations. The ToM module performs this inference through

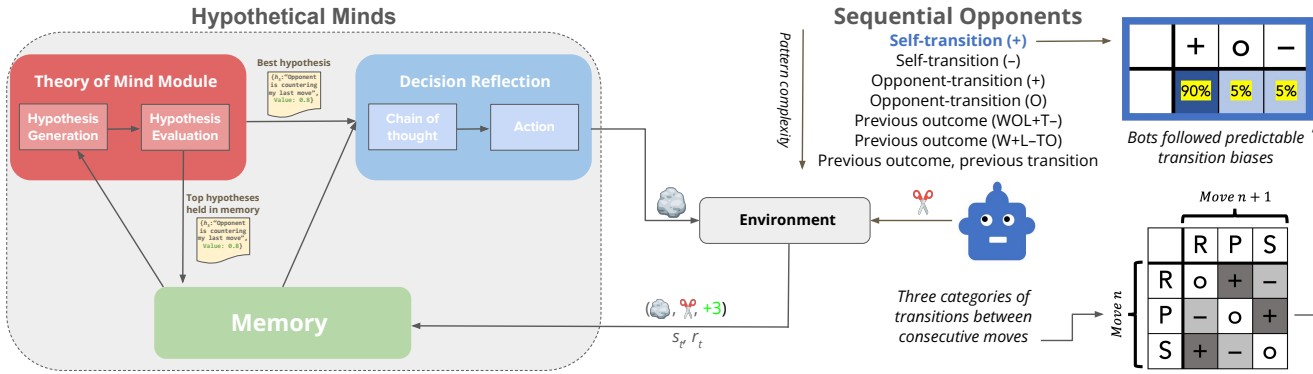

Figure 1: Architecture of Hypothetical Minds and Sequential algorithmic opponents.

two components: **Hypothesis Generation** where we prompt the LLM to generate natural language hypotheses representing beliefs about the latent variables $h_i = p(\Theta)$, and **Hypothesis Evaluation** that scores each hypothesis based on how accurately the hypothesis predicts future behavior. The model selects the best hypothesis $h^*$ by approximating the Maximum a Posteriori (MAP) estimation:

$$h^* = \arg\max_{h_i \in \mathcal{H}} p(h_i|\mathbf{a}) = \arg\max_{h_i \in \mathcal{H}} \frac{p(\mathbf{a}|h_i)p(h_i)}{p(\mathbf{a})}$$

where $p(\mathbf{a}|h_i)$ is the likelihood of observing action sequence $\mathbf{a} = [a_1, a_2, ..., a_t]$ given hypothesis $h_i$, $p(h_i)$ is the prior probability of hypothesis $h_i$, and $p(\mathbf{a})$ is the marginal probability of the observed actions (constant across hypotheses).

**Hypothesis Generation** uses an LLM to produce hypotheses about opponents' strategies (e.g., "opponent counters my previous move") based on memory of past interactions. This process implicitly incorporates prior knowledge $p(h_i)$ through both the background knowledge in the LLM's pretrained weights, and a refinement mechanism that presents top-valued hypotheses to the LLM when generating new ones.

**Hypothesis Evaluation** approximates the likelihood function $p(\mathbf{a}|h_i)$ by scoring hypotheses based on prediction accuracy. For each round, the model: 1. Selects the top 5 hypotheses with highest current values (where `top_h` is a configurable hyperparameter), 2. queries the LLM to predict the opponent's next move given each hypothesis and 3. updates hypothesis values after observing the opponent's actual move. Hypotheses are valued by calculating recency-weighted prediction accuracy using a reinforcement learning update rule since agents can change their strategies in dynamic settings (Rescorla, 1972; Daw & Tobler, 2014). Let hypothesis $h_i$ generate a prediction $\hat{a}_i$ for the opponent's next move. After observing the actual move $a$, we calculate an intrinsic reward:

$$r_i = \begin{cases} +1 & \text{if } \hat{a}_i = a \\ -1 & \text{if } \hat{a}_i \neq a \end{cases}$$

The value of hypothesis $h_i$ is then updated via reward pre-

diction errors:

$$V_{h_i} \leftarrow V_{h_i} + \alpha \cdot (r_i - V_{h_i})$$

where $\alpha = 0.3$ is the learning rate that determines how much recent predictions influence the hypothesis value. This update rule ensures that values remain bounded in [-1, 1], with repeated successful predictions pushing values toward 1.

A hypothesis becomes "validated" when its score exceeds threshold $V_{thr} = 0.7$, a value typically obtained after three consecutive correct predictions or 85% prediction accuracy. This ensures that the model only considers reasonably accurate hypotheses; however, performance is not sensitive to this value (Cross et al., 2024). Once validated, the hypothesis is used for decision-making without generating new hypotheses until its score falls below threshold. If no hypothesis meets the threshold, the last generated hypothesis is used by default.

This approach allows the ToM module to maintain and evaluate multiple competing hypotheses about other agents' strategies, select the most probable explanation for observed behavior, and adapt as agents' strategies change over time.

The **Decision Reflection** component takes the best hypothesis (either the validated hypothesis or the most recently generated one) and uses chain-of-thought reasoning (Wei et al., 2022) to determine what it thinks its opponent will play and determines what the optimal response should be (see prompts in Supplementary Material). This component conditions its reasoning on both the memory of past interactions and the selected hypothesis about the opponent's strategy. Finally, this component prompts the LLM to generate its next move (rock, paper, or scissors) based on its chain-of-thought reflection (Yao et al., 2022). This prompt is also used in hypothesis validation to elicit predictions of the opponent's next move for counterfactual hypotheses not used in that trial.

## Results

### Experiment 1: Hypothetical Minds Reproduces Human Performance Patterns

First, we characterize human performance patterns in this RPS task, which will serve as the empirical foundation for

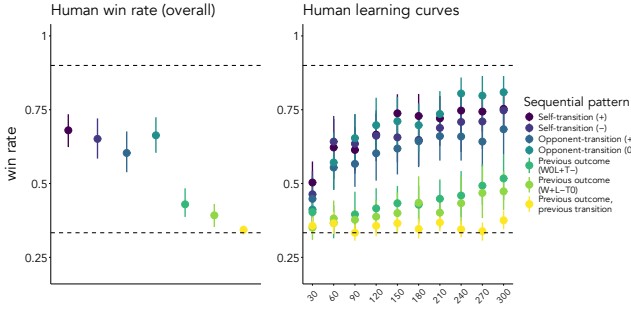

Figure 2: Human performance against sequential opponents. Error bars represent bootstrapped 95% confidence intervals.

our cognitive modeling. Humans demonstrated a clear pattern of success in detecting and exploiting sequential patterns, with performance varying systematically with pattern complexity. Against bots using simple transition rules, participants achieved strong performance, with win rates ranging from 57.9-66.5%. These rates significantly exceeded chance levels (33%) across all four transition-based strategies (all p < .001). Learning trajectories reinforced this pattern. With transition-based bots, participants quickly discovered and exploited the underlying patterns, maintaining high performance throughout most trials.

Meanwhile, performance was dramatically worse against more complex opponents. Against bots that conditioned their transitions on previous game outcomes, participants showed only modest success, with win rates of 41.3% and 39.7% for the two outcome-dependent strategies. While still statistically above chance (p < .005), these win rates were substantially lower than those achieved against the simpler transition-based opponents. Finally, participants failed to perform above chance against the most complex bot, which combined outcome and previous transition dependencies (34.1%, p = .30).

Having characterized human performance patterns, we next examine whether Hypothetical Minds can capture these patterns through comparison with baseline LLM architectures and a GPT-4o backbone. We include a set of baselines/ablations of our model (see Figure 8 in Supplement for depictions of architectures):

- **No Hypothesis Evaluation**. We ablate hypothesis evaluation by not computing values for previously generated hypotheses; therefore a hypothesis cannot be validated and subsequently used to accurately anticipate the opponent's next move. Also, we do not show high-valued hypotheses in the prompt such that the LLM can refine good guesses. Thus, every round the agent generates a new hypothesis from scratch and uses it for planning its next move.
- **ReAct** produces chain-of-thought (CoT) reasoning before asking the LLM to take an action to separate reasoning from acting ([Yao et al., 2022]). In this context, this corresponds to an ablation of the ToM module (no hypothesis generation or evaluation). With solely the decision reflection, the agent is

prompted to use chain-of-thought reasoning to think about its next move given the history of past rounds.

- **Base LLM**. Prompts the LLM to play without any CoT, ablating the decision reflection.

The Hypothetical Minds model corresponds closely with human performance across 6/7 opponent types (Figure 3), achieving the closest set of win rates to human participants in $L^2$ distance (0.527) compared to baselines (Table 2; No Hypothesis Evaluation: 0.736, ReAct: 0.936, and Base LLM: 0.911). Like humans, it achieves high win rates (74-77%) against simple transition-dependent strategies while showing similar limitations with the more complex patterns. Both humans and our model achieved only modest success above chance against bots using outcome-dependent strategies, while performance dropped to chance levels against the most complex bot combining outcome and transition dependencies.

We also conducted a comparison of learning trajectories. In addition to similarity of episode-level win-rates, Table 2 presents two learning metrics: trajectory $L^2$ distance (capturing the similarity of win rates across time in bins of 10 rounds), and trajectory correlation (measuring the directional alignment of learning trends). HM maintained significantly better alignment with human learning patterns (trajectory $L^2 = 2.95$) than alternatives. HM also showed positive trajectory correlation with human data ($r = 0.38$) whereas baselines without proper hypothesis evaluation or theory of mind components showed negative or near-zero correlations. This correspondence between human and model performance—spanning both successes, failures, and learning curves—suggests Hypothetical Minds may capture key aspects of the cognitive processes and limitations underlying human pattern recognition in this task.

The model comparison reveals what is necessary for human-like sequential pattern recognition. The full Hypothetical Minds architecture integrates four key components: memory of past rounds, a Theory of Mind module (with hypothesis generation and evaluation subcomponents), decision reflection, and action selection. Our analyses demonstrate that the Theory of Mind module is particularly crucial—the ReAct baseline, which removes this module while maintaining memory and decision reflection, has win rates of only 14-50% against simple transition-based opponents. An even simpler baseline that directly generates moves without CoT (GPT-4o Base LLM) performs at or below chance levels for all but one of the bots. With hypothesis evaluation ablated, the agent no longer has memory of past guesses and how predictive those guesses are about the opponent's behavior; it must generate the correct hypothesis anew on every round. This significantly reduces performance when the correct hypothesis could be discovered through hypothesis evaluation.

An intriguing anomaly emerges with the Opponent-transition$(0)$ bot, which simply copies the player's previous move. While humans readily detect and exploit this pattern, the Hypothetical Minds model with a GPT-4o base LLM struggles. The model frequently generates incorrect hypotheses that lead to below-chance performance, often misinterpreting the copying behavior as

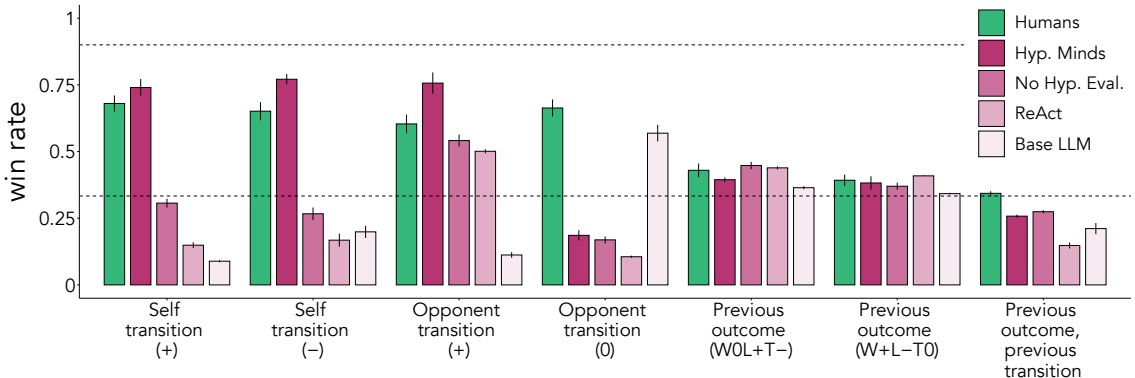

Figure 3: HM baseline model performance. Error bars reflect SEM over 3 seeds.

a strategy like "playing the move that beats my previous play" (Opponent-transition$(+)$). Surprisingly, the base LLM without chain-of-thought is the only variant that performs above chance against this opponent. This suggests that the structured reasoning processes we've implemented—particularly the chain-of-thought prompting—actually impairs performance on this specific pattern with GPT-4o. However, when using Llama 3 as the base model, Hypothetical Minds performs well above chance on this opponent (Figure 4). This indicates that our architecture can potentially achieve strong performance on all opponents that humans readily exploit, though this capability depends on the underlying LLM.

Figure 4 depicts how the choice of base language model affects performance by comparing GPT-4o, Llama 3, and GPT-3.5 as backbones to our cognitive architecture. GPT-4o demonstrated the strongest overall performance and closest alignment with human behavior patterns, achieving similar win rates to humans across most opponents. However, the performance patterns of different language models exhibited interesting variation. Llama 3 showed surprisingly strong performance against certain opponents, particularly excelling at exploiting the Opponent-transition$(0)$ strategy and achieving above-chance performance against Previous outcome$(W + L - T0)$. Yet it struggled with self-transition patterns and more complex strategies, performing at chance levels like humans. GPT-3.5 showed the most limited capabilities, only achieving notable success against the Opponent-transition$(+)$ strategy. This model also frequently responded with formatting errors. Thus, the reasoning capabilities of the base model matter; more powerful LLMs demonstrate better pattern recognition. We use GPT-4o for the remaining experiments.

For the human data, each participant was asked to describe their opponent's strategy after completing all 300 rounds of play. We conducted a qualitative analysis of these responses along with HM's hypotheses, categorizing them according to whether they correctly identified the underlying sequential dependency type (Self-transition, Opponent-transition) or proposed an alternative out-of-distribution explanation. Table 3 demonstrates some similarities between human and HM hy-

potheses, with both identifying similar pattern types for simple bots and making comparable misattributions by sometimes inferring random or static strategies when unable to detect the true underlying pattern. Both humans and HM also produce out-of-distribution hypotheses that are not easily categorizable or compressible due to the expressivity of language.

### Experiment 2: Is hypothesis search the bottleneck?

Having established that both humans and HM struggle with complex sequential patterns, we next conducted an experiment to isolate whether the limitation lies in hypothesis generation, hypothesis evaluation, or move selection. It is possible that the poor performance stems from difficulty in producing hypotheses, which requires intuitions aligned to the task. On the other hand, it may stem from the cognitive demands required to evaluate hypotheses in light of evidence from the opponent's moves. Or, humans and the model may just struggle to exploit the strategies even when they are known. To dissociate these possibilities, we augment the model with perfect knowledge: instead of having to generate and evaluate hypotheses about opponent behavior, the model receives an explicit description of the true strategy it faces.

This **Give Hypothesis** augmentation represents the opposite of an ablation—rather than removing model components to test their necessity, we replace the Theory of Mind module with oracle knowledge (see Figure 9 in Supplement for depictions of architecture). If the model can successfully exploit opponent patterns when given their true description in natural language, this would suggest that hypothesis search, not strategy implementation, is the key cognitive bottleneck. Conversely, if the model continues to struggle even with perfect knowledge, this would indicate fundamental limitations in the ability to exploit complex sequential strategies, regardless of how they are discovered.

We also created an intermediate augmentation model called **Choose Hypothesis**, which provides the agent with a list of all seven possible hypotheses (Figure 9). The model then runs hypothesis evaluation as usual, querying the LLM conditioned on each hypothesis to predict the next action and scoring them accordingly, with the best hypothesis sent to the

Table 1: Model-Human Similarity

| Metric | Human-Human | Hyp. Minds | No Hyp. Eval. | ReAct | Base LLM |
|---|---|---|---|---|---|
| Win Rate $L^2$ | 0.17 | **0.53** | 0.74 | 0.94 | 0.91 |
| Trial bins $L^2$ | 1.40 | **2.95** | 4.27 | 5.17 | 4.96 |
| Trial bins $r$ | 0.81 | **0.38** | -0.23 | -0.27 | -0.02 |

Table 2: Similarity between win rates and trajectories of win rates by time (in bins of 10 trials). Human-Human comparison, the noise ceiling, computed by splitting participants into two groups and computing similarity between groups.

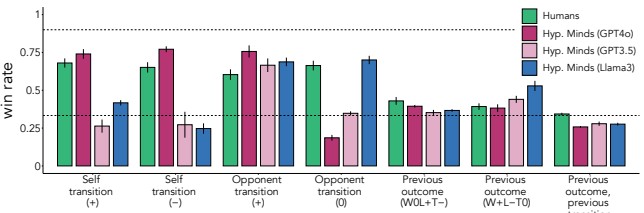

Figure 4: LLM comparison

decision reflection component. This tests whether the bottleneck is in generating plausible hypotheses or in correctly evaluating them.

When provided with oracle knowledge of opponent strategies, model performance improved dramatically for most opponents, reaching win rates above 80% against both transition-based and outcome-dependent bots. This improvement was particularly notable for the outcome-dependent strategies, where performance jumped from near-chance levels to consistently high win rates. This suggests that for these moderately complex patterns, the primary bottleneck lies in hypothesis search rather than strategy implementation—once the pattern is known, the model can effectively exploit it.

The "Choose Hypothesis" model performed nearly as well as the "Give Hypothesis" oracle model, though with slightly lower performance overall (Figure 5). Both showed substantial improvements over the default Hypothetical Minds model against the Opponent-transition($0$) bot and the two previous outcome bots, while performance was largely similar against the transition bots (where HM was already successful). This result suggests that hypothesis generation is the main cognitive bottleneck for the outcome based opponents; the agent is given knowledge of the possible strategies or the true strategy, it quickly converges on the correct hypothesis.

However, both augmentation models continued to struggle with the most complex opponent with previous outcome & transition dependencies, even when given its true strategy description. This strategy conditions moves on both the previous outcome and the opponent's previous transition type, requiring tracking of multiple contingencies across three time steps. The strategy's complexity is evident in its natural language description when given to the LLM agent, which requires nine distinct condition-action pairs (compared to three pairs for the outcome-dependent strategies) totaling 352 words (see prompts in Supplementary Material). The model's continued difficulty even with perfect knowledge suggests that for highly complex sequential patterns, there are additional bottlenecks beyond hypothesis generation, namely, the cognitive demands of accurately reasoning about and implementing responses to multiple nested contingencies.

## Experiment 3: Improving hypothesis generation

**Increasing the number of hypotheses evaluated doesn't improve performance** Having identified hypothesis gener-

ation as the primary cognitive bottleneck, we next investigated potential mechanisms for improving this process. One straightforward possibility is that considering more candidate hypotheses increases the likelihood of discovering the true opponent strategy.

To test this, we manipulated the number of hypotheses maintained and evaluated by the model. While the model generates a new hypothesis after each round (299 total across 300 rounds), computational constraints require us to limit how many hypotheses are actively used for prediction and evaluation. By default, only the top 5 highest-valued hypotheses are maintained. We systematically varied this parameter (top_h) to examine its effect on performance.

Results revealed that the critical factor was maintaining any hypotheses at all, rather than the specific number maintained (Figure 6 Top). Performance improved dramatically when moving from top_h=0 (no hypothesis evaluation, equivalent to the baseline above) to top_h=1, but showed minimal gains with further increases. This suggests that the key benefit comes from having some memory of past hypotheses, allowing the model to retain and build upon promising explanations while naturally discarding poor ones (Bramley et al., 2017; Zhao et al., 2024; Fränken et al., 2022). Increasing the hypothesis pool beyond 5 showed no additional benefit, with performance remaining relatively flat as top_h increased from 1 to 9. This indicates that the challenge in hypothesis generation lies not in considering more possibilities, but in generating the right kinds of hypotheses in the first place.

**Wider hypothesis search doesn't help** If considering more hypotheses doesn't improve performance, there are two possible explanations: either the model is searching in the wrong region of hypothesis space, or its search distribution is too narrow. To differentiate between these, we tested whether widening the search distribution would help by increasing the temperature parameter in our LLM from T=0.2 to T=1.0, which increases the randomness of generated hypotheses.

This manipulation degraded performance against all five opponents where Hypothetical Minds originally performed above chance, while showing no improvement against the opponents it struggles with (Figure 6 Bottom). Additionally, with a higher temperature, the responses are more prone to formatting errors and low quality responses. This result suggests that the challenge isn't simply one of widening the distribution

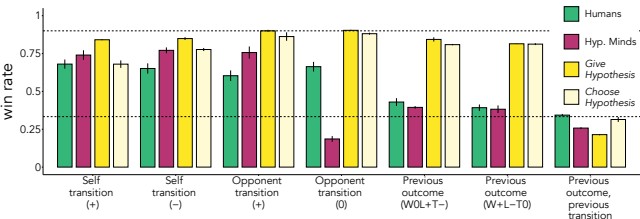

Figure 5: Augmentations for hypothesis generation and hypothesis evaluation

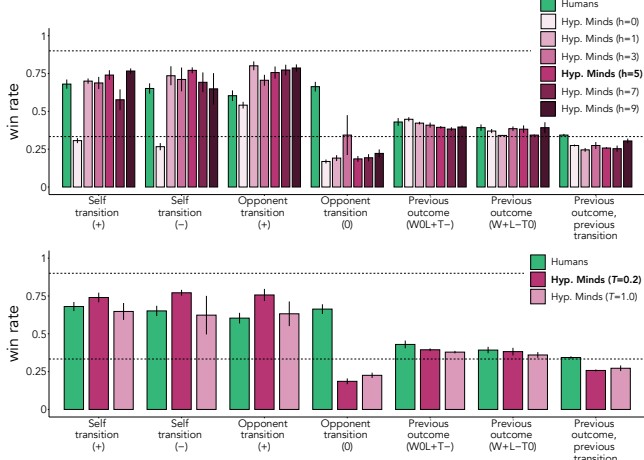

Figure 6: Varied number of hypotheses and LLM temperature

that the model samples hypotheses from. Rather, the model appears to be searching in fundamentally incorrect regions of hypothesis space. The prior knowledge encoded in the LLM—shaped by its training on natural language data—may not include the kinds of algorithmic strategies employed by our artificial opponents, particularly for the more complex patterns. Similarly, human participants may struggle because these patterns fall outside their prior expectations about strategic behavior. The failure of this wider search suggests that the fundamental bottleneck lies in having appropriate priors about possible behavioral patterns.

**Verbal Scaffolding Aids Hypothesis Generation**  If the challenge lies in having appropriate priors about possible behavioral patterns, then restructuring how these patterns are represented might help align the search space with the true strategies. This intuition aligns with extensive research on learning and pedagogy showing that humans benefit from scaffolding—structured guidance that helps direct attention to relevant features without explicitly providing answers (Vygotsky, 1978; Wood et al., 1976; Belland, 2014). Just as a teacher might help students notice patterns in math problems by highlighting key relationships, we investigated whether providing verbal scaffolding could guide the model's hypothesis search toward more relevant regions of strategy space.

Specifically, we implemented two theoretically-grounded scaffolding approaches based on cognitive theories of guided learning. First, we developed **Attention Scaffold** prompts that highlight relevant contingencies without revealing the specific pattern. For example: *"There are three different kinds of transitions a player can make from their last round's move to their current move: (defines transitions) Pay particular attention to whether your opponent's transition type varies depending on the previous outcome (win/loss/tie)."*

Second, we tested **Analogical Scaffold** prompts that provide a structurally similar but non-identical example with a matched level of abstraction, similar to parallel problems (Gentner & Hoyos, 2017; Lin & Singh, 2011; Singh, 2008). These one-shot examples used different contingencies from those tested to avoid answer provision while activating the relevant schema. For instance, when testing against a Previous outcome($W0L + T-$) bot, we provided this example: *"After a win the opponent plays the move that would lose*

*to their last round's move. After a loss the opponent plays the same move as they did in the last round. After a tie the opponent plays the move that would beat their last round's move."*

Figure 7 demonstrates that **Attention Scaffold** hints significantly improve performance against the outcome-dependent bots that were previously bottlenecked by hypothesis generation, with win rates increasing to 67% and 72%. This scaffolding also enhanced performance less dramatically against the Opponent-transition($0$) bot to 47%. Performance remained consistently high for the three opponent types that Hypothetical Minds already exploited effectively. Providing **Analogical Scaffold** information also improved performance, though less dramatically. Win rates against the outcome-dependent bots increased to 47% and 58%, and to 38% for Opponent-transition($0$). As with attention-directing hints, performance remained robust for the other transition bots. These results suggest that the hypothesis space is successfully redirected both by attention to relevant contingencies and by the availability of appropriate abstract schemas for organizing sequential information. The differential effectiveness of the two scaffolding approaches further indicates that making contingencies salient may be more crucial than providing structural analogies when the challenge involves detecting complex dependencies in sequential decision-making. This result introduces an intriguing hypothesis about cognition and learning that can then be tested in humans: that verbal scaffolding which directs attention to relevant contingencies may help humans overcome the same pattern recognition limitations observed in our cognitive model.

## Discussion

Our findings demonstrate that Hypothetical Minds (HM) reproduces key features of human performance in iterated games of rock, paper, scissors, succeeding on simple transition patterns while struggling with more complex dependencies. This correspondence between human and model behavior—spanning both successes and failures—suggests that

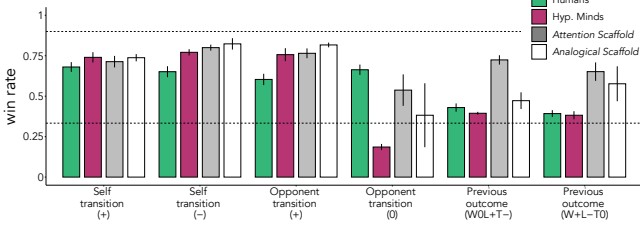

Figure 7: Result of scaffolding interventions on hypothesis generation

HM can model the cognitive processes underlying how people detect and exploit patterns in sequential decision-making.

However, our finding that HM with GPT-4o struggles with the Opponent-transition$(0)$ bot—which simply copies its opponent's previous move—highlights an important limitation for LLMs as cognitive models. While humans readily detect the Opponent-transition$(0)$ pattern, the model frequently misinterprets this copying behavior, leading to below-chance performance. This discrepancy may stem from differences in representational space between humans and language models. Interestingly, Llama 3 performed well on this pattern, suggesting the limitation relates to properties of the base LLM rather than our cognitive architecture. The behavioral differences across LLM base models may arise from a combination of factors ranging from pretraining data exposure, posttraining recipes, and architectural differences. Also, LLMs exhibit sensitivity to prompt structure and information presentation (Webson & Pavlick, 2021; Loya et al., 2023; Sclar et al., 2023), sometimes leading to unexpected failures on seemingly simple tasks despite capturing many aspects of human reasoning.

Our quantitative and qualitative analysis of hypotheses highlight the ways in which prior knowledge and inductive biases constrain adaptive learning in both artificial and human cognition (Binz et al., 2024). Humans exhibit strong priors about sequential patterns and often fail to infer structures that deviate from familiar cognitive schemas (Kahn et al., 2018; Acuna & Schrater, 2008). LLMs are also not fully flexible in-context learners (Si et al., 2023), as we see here with HM's failure to exploit complex bots. Our augmentation and scaffolding results point to promising pedagogical approaches to explore with humans to see if giving strategies or hints similarly improves performance for moderately complex bots or even all bots. Our results suggest that under time-constraints humans may be similarly bottlenecked when faced with the most complex bot due to its lengthy list of dependencies and bounded rationality (Simon, 1972; Lieder & Griffiths, 2020). However, this needs to be verified and we hypothesize that many humans would be able to perform well when given the hypothesis (and maybe without it) with external tools, higher motivation, and more time.

The emergence of test-time scaling offers novel opportunities to evaluate scientific hypotheses about resource rational-ity with meta chain-of-thought (V. Xiang et al., 2025; Guo et al., 2025; Muennighoff et al., 2025), test-time verification (Wang et al., 2023; Yao et al., 2023), and program synthesis/tool use (Wong et al., 2023; Li et al., 2025; Kang et al., 2025). HM implements a form of test-time scaling, similar to best-of-N selection (Nakano et al., 2021) with verification implemented by the hypothesis evaluation mechanism. However, our results suggest that this scaling is limited at the order of magnitude tested, as the hypotheses are generated from a narrow distribution. The refinement method could potentially worsen diversity by concentrating search around already-explored hypotheses. Humans similarly tend to update their hypotheses locally (Bramley et al., 2017; Zhao et al., 2024; Bramley & Xu, 2023; Fränken et al., 2022). Future work should explore this tradeoff more explicitly and use curiosity approaches to increase the diversity of LLM responses (Poesia et al., 2024).

The computational approach taken with HM is distinct from prior modeling work in sequential decision making paradigms like rock, paper, scissors (Sepahvand et al., 2014; Rapoport & Budescu, 1997; West & Lebiere, 2001), yet HM's cognitive process has analogs in other domains of computational modeling. For instance, in some learning contexts, both animals and humans may generate an unbounded number of possible latent causes of observed data until a satisfactory explanation is found, a phenomenon that has been modeled using the Chinese restaurant process (Gershman et al., 2015). This approach explains a range of learning behaviors, including categorization and Pavlovian conditioning (Anderson, 1991; Sanborn et al., 2010; Gershman & Niv, 2013). HM likewise generates and refines hypotheses about the opponent's strategy until one aligns with observed data. Similarly, Bayesian inverse planning models propose that humans infer others' beliefs and goals through a structured probabilistic process that accounts for a broad array of Theory of Mind capabilities (Baker et al., 2009, 2017; Rusch et al., 2020). Inspired by these approaches, our model also infers the most likely intentions that explain an agent's actions, while leveraging the flexibility and domain generality of an LLM's representational space to represent a potentially unbounded list of hypotheses.

The results add to emerging evidence that LLMs can operate as cognitive models, producing accurate task representations and outperforming traditional task-specific models (Binz & Schulz, 2023). Similarly, by integrating in-depth interviews with human participants, LLM-based models can replicate human responses in surveys and economic games with striking fidelity (Park et al., 2024). However, validating LLM-based cognitive models remains a significant challenge. Recent efforts have developed more rigorous validation frameworks to assess the alignment between model-generated behaviors and human cognition (Vezhnevets et al., 2023). Here, our primary validation approach involved comparing different LLM agent architectures to identify the components necessary for producing human-like behavioral patterns.

## Acknowledgements

This work was supported by the following awards: E.B. is supported by NSF SBE Postdoctoral Research Fellowship 2404706, T.G. is supported by grants from Stanford's Human-Centered Artificial Intelligence Institute (HAI) and Cooperative AI, J.E.F. is supported by NSF DRL award 2400471 and NSF CAREER award 2047191 and an ONR Science of Autonomy award. To D.L.K.Y.: Simons Foundation grant 543061, NSF CAREER grant 1844724, NSF Grant NCS-FR 2123963, Office of Naval Research grant S5122, ONR MURI 00010802, ONR MURI S5847, and ONR MURI 1141386–493027. N.H. is supported by NSF Grant No. 2302701, and by the Stanford HAI Hoffman-Yee Research Grant and funding from the Stanford Graduate School of Education.

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

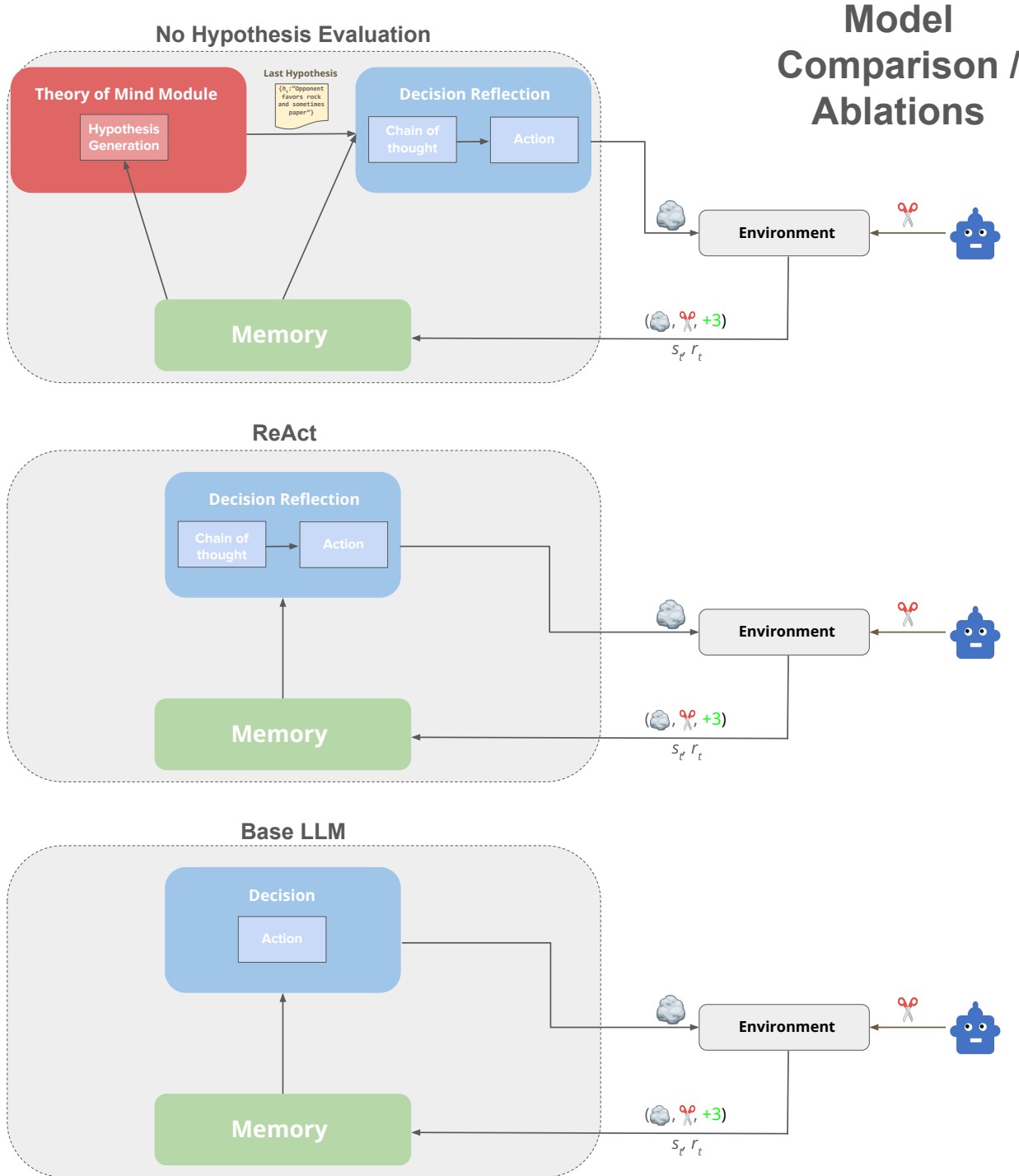

Figure 8: LLM architectures in model comparison.

# Model Augmentations

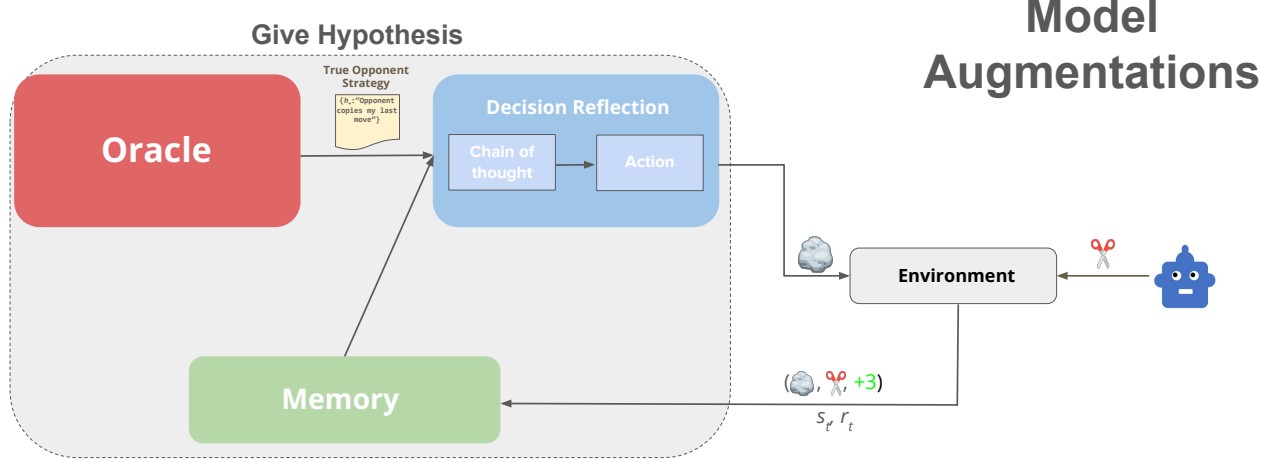

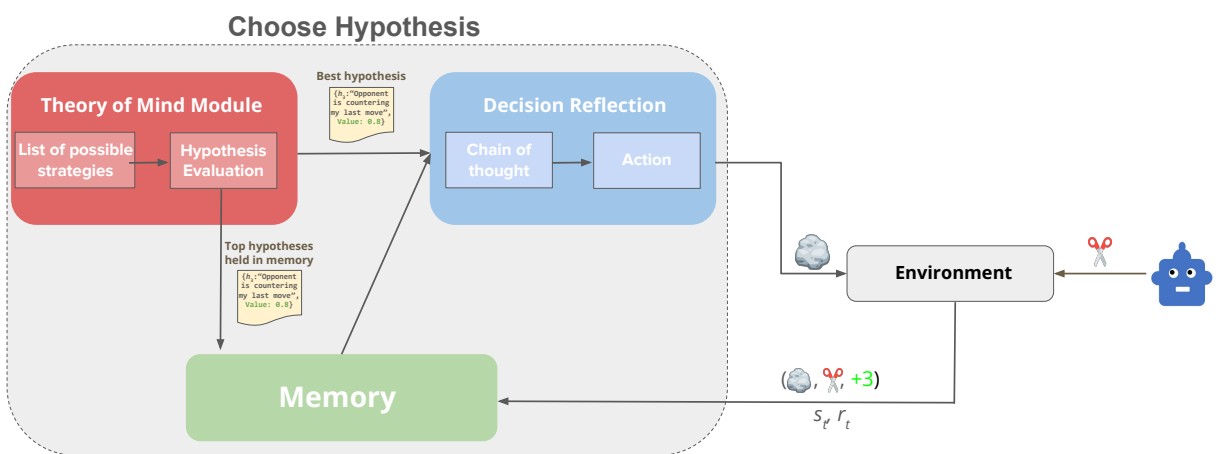

Figure 9: LLM architectures in model augmentations.

| Human Hypotheses | Strategy Type |
|---|---|
| After a while I realized that if I picked rock, paper, scissors in order I would either win the round or tie. | **Self-transition** |
| rotating through rock paper and scissors when i saw a loop | **Self-transition** |
| Initially I clicked randomly. Then, it seemed like my opponent was copying whatever my last move was, so I cycled between all three choices in the same order. I noticed that this work 100% of the time, occasionally the opponent would pick something else. However, it worked often enough that it was by far the best strategy. Perhaps there was some way of predicting when it would change to something besides what I just played, but I didn't care enough to find out. I think I had 600+ points, and the opponent had -110 or less points by the end | **Opponent-transition** |
| By selecting the same option mostly | None (static bot) |
| I didn't really have one. It seemed like the computer was randomly choosing options. | None (random) |
| I would push on scissors multiple times so that we would match and then push on rock and most of the time, my opponent would still be on scissors. Other than that, I just tried to stay random. | None |
| **Hypothetical Minds Hypotheses** | **Strategy Type** |
| I think my opponent is cycling through paper, rock, and scissors, with occasional deviations towards scissors | **Self-transition** |
| The opponent is cycling through scissors, paper, and rock with occasional deviations by repeating the last move | **Self-transition** |
| The opponent follows a reactive strategy, countering my last move with the move that would beat it, with a preference for paper, especially after I play rock or scissors. They occasionally mirror my move, leading to ties. | **Opponent-transition** |
| Based on the interaction history, I notice that my opponent has played paper a significant number of times, especially in the recent rounds. In fact, in the last 10 rounds, they have played paper 7 times. This suggests that they may be playing a static bias strategy, favoring paper over rock and scissors. | None (static bot) |
| I think my opponent is playing randomly with equal probability for rock, paper, and scissors | None (random) |
| The opponent favors playing paper more frequently, with occasional switches to scissors. They might be using a simple alternating strategy between paper and scissors, with a slight bias towards paper | None |

Table 3: Comparison of Human and Hypothetical Minds Hypotheses. **Bold** strategies are in-distribution of the test set of opponent strategies. Out-of-distribution (OOD) simple strategies with clear labels are labeled with None (), and OOD complex or other strategies labeled None.

## Hypothesis Generation Prompt

An interaction with the other player has occurred at round `step`, {`last_round_info`}.
The total interaction history is: {`interaction_history`}.
Here are your previous hypotheses about the algorithm your opponent is playing:
{`top_hypotheses`}.
What is your opponent's likely policy given their plays? Think step by step about this given the interaction history.
If your previous hypotheses are useful, you can iterate and refine them to get a better explanation of the data observed so far.
If a hypothesis already explains the data very well, then repeat the hypothesis in this response.
They may be playing the same static policy every time, a complex strategy to counter you, or anything in between.
They are not necessarily a smart agent that adapts to your strategy; you are just playing an algorithm.
Are you getting positive or negative reward when playing the same choice?
For example, getting positive reward every time you play rock.
If so, your opponent may be playing a static strategy, and you can exploit this by playing the counter strategy.
Once you have output a hypothesis about your opponent's strategy with step-by-step reasoning, you can use the hypothesis to inform your strategy.
In the second part of your response, summarize your hypothesis in a concise message following Python dictionary format, parsable by `ast.literal_eval()`, starting with:
Example summary:

```
{
    'Opponent_strategy': ''
}
```

This summary will be shown to you in the future to help you select the appropriate counter-strategy.
You will be prompted again shortly to select your next play, so do not include that in your response yet.

## Decision Reflection Prompt

An interaction with the other player has occurred at round {`step`}, {`self.interaction_history[-1]`}.
The total interaction history is: {`self.interaction_history`}.
You last played: {`self.interaction_history[-1]['my_play']`}
You previously guessed that their policy or strategy is: {`possible_opponent_strategy`}.
High-level strategy Request:
Provide the next high-level strategy for player {`self.agent_id`}.
Think step by step in parts 1 and 2 about which strategy to select based on the entire interaction history in the following format:
1. 'predicted_opponent_next_play': Given the above mentioned guess about the opponent's policy/strategy, and the last action you played (if their strategy is adaptive, it may not be), what is their likely play in the next round.
2. 'my_next_play': Given the opponent's likely play in the next round, what should your next play be to counter this?
3. In the 3rd part of your response, output the predicted opponent's next play and your next play as either 'rock', 'paper', or 'scissors' (use no other string) in following Python dictionary format, parsable by `ast.literal_eval()` starting with:
Example response:
1. 'predicted_opponent_next_play': Given that my opponent is playing a rock policy, I believe their next play will be a rock.
2. 'my_next_play': Given that my opponent is playing a rock policy, I believe my next play should be paper.

```
{
  'predicted_opponent_next_play': 'rock',
  'my_next_play': 'paper'
}
```

## Give Hypothesis

### Hypothesis Mappings

**self_transition_up:** *The opponent plays the move that would beat their last rounds move*

**self_transition_down:** *The opponent plays the move that would lose to their last rounds move*

**opponent_transition_up:** *The opponent plays the move that would beat their opponents last rounds move*

**opponent_transition_stay:** *The opponent plays the same move as their opponents last rounds move*

**W_stay_L_up_T_down:** *After a win the opponent plays the same move as they did in the last round. After a loss the opponent plays the move that would beat their last rounds move. After a tie the opponent plays the move that would lose to their last rounds move*

**W_up_L_down_T_stay:** *After a win the opponent plays the move that would beat their last rounds move. After a loss the opponent plays the move that would lose to their last rounds move. After a tie the opponent plays the same move as they did in the last round*

**prev_outcome_prev_transition:** *The opponents transition from one round to the next depends on both the previous outcome (win, lose, or tie) and the previous transition the opponent made.*

- *After a win in which the opponent played the move in the last round that would beat the opponent's move two rounds ago, the opponent plays the move that would beat their last rounds move.*
- *After a win in which the opponent played the move in the last round that would lose to the opponent's move two rounds ago, the opponent plays the move that would lose to their last rounds move.*
- *After a win in which the opponent played the same move in the last round as the opponent played two rounds ago, the opponent plays the same move as they did in the last round.*
- *After a loss in which the opponent played the move in the last round that would beat the opponent's move two rounds ago, the opponent plays the same move as they did in the last round.*
- *After a loss in which the opponent played the move in the last round that would lose to the opponent's move two rounds ago, the opponent plays the move that would beat their last rounds move.*
- *After a loss in which the opponent played the same move in the last round as the opponent played two rounds ago, the opponent plays the move that would lose to their last rounds move.*
- *After a tie in which the opponent played the move in the last round that would beat the opponent's move two rounds ago, the opponent plays the move that would lose to their last rounds move.*
- *After a tie in which the opponent played the move in the last round that would lose to the opponent's move two rounds ago, the opponent plays the same move as they did in the last round.*
- *After a tie in which the opponent played the same move in the last round as the opponent played two rounds ago, the opponent plays the move that would beat their last rounds move.*

# Attention Scaffold

## Self-Transition Strategies

**self_transition_up and self_transition_down:**
There are three different kinds of transitions a player can make from their last round's move to their current move.

- An up transition occurs when they play the move that would beat their last round's move.

- A down transition occurs when they play the move that would lose to their last round's move.

- A stay transition occurs when they play the move that is the same as their last round's move.

## Opponent-Transition Strategies

**opponent_transition_up and opponent_transition_stay:**
There are three different kinds of transitions a player can make from your last round's move to their current move.

- An up transition occurs when they play the move that would beat your last round's move.

- A down transition occurs when they play the move that would lose to your last round's move.

- A stay transition occurs when they play the move that is the same as your last round's move.

**Outcome-Based Strategies**

**W_stay_L_up_T_down and W_up_L_down_T_stay:**
There are three different kinds of transitions a player can make from their last round's move to their current move.

• An up transition occurs when they play the move that would beat their last round's move.

• A down transition occurs when they play the move that would lose to their last round's move.

• A stay transition occurs when they play the move that is the same as their last round's move.

Pay attention to the type of transitions your opponent makes after a win, a loss, and a tie.

# Analogical Scaffold Examples

**self_transition_up and self_transition_down:** "The opponent plays the move that would tie to their last rounds move."

**opponent_transition_up and opponent_transition_stay:** "The opponent plays the move that would lose to your last rounds move."

**W_stay_L_up_T_down and W_up_L_down_T_stay:** "After a win the opponent plays the move that would lose to their last rounds move. After a loss the opponent plays the same move as they did in the last round. After a tie the opponent plays the move that would beat their last rounds move."

