# OpenReview forum: "Understanding Human Limits in Pattern Recognition: A Computational Model of Sequential Reasoning in Rock, Paper, Scissors"
_ccneuro.org/CCN/2025/Proceedings — CCN 2025 Proceedings asProceedingsPoster_

### Official Review · Reviewer_R5Wg · 2025-03-23
**a well-executed study using language models & decision making in repeated Rock Paper Scissors, with insightful analysis on where performance bottlenecks emerge**

**Soundness:** 3
**Clarity:** 3

**Comments:**

This paper modeled how people make decisions in repeated Rock Paper Scissors games. Working with an empirical dataset where people played with seven bots - each programmed with a strategy and overall ranged over three complexity levels- this paper examined a computational model Hypothetical Mind (HM). HM has three components: (1) memory, storing playing histories, (2) strategic reasoning, or theory of mind, hypothesizing the strategy that the opponent (the bot) might be using, and (3) decision-making, choosing a move based on the inferred strategy. The authors leveraged large language models to propose hypotheses. Each hypothesis was then evaluated (scored), and the preferred hypothesis was fed to the decision-making module. Overall, HM matched people’s performance, with some variations across the language model of choice. The authors also conducted comprehensive analysis including ablation and augmentation studies, and found that the major bottleneck of performance likely comes from hypothesis generation - when the underlying strategy is complex, language models tend to sample hypotheses in the wrong direction, where verbal scaffolding can be helpful to some extent.

This study is well designed, conducted, analyzed, and reported. This work serves as a great example enhancing traditional decision-making models with LLMs in principled ways, and would be of great interest to the community of CCN. The paper is well written and easy to follow, making it possible to take similar approaches in other decision-making tasks. The results are carefully analyzed - especially the augmentation study is very informative of showcasing where the performance gap lies. The analysis followed a logical order and it was satisfying to read.

I would appreciate it if the authors could offer some example strategies that LLMs inferred, and possibly statistics about how many of the LLMs inferred strategies aligned with the ground truth. Additionally, are there patterns of how LLMs sample the next hypothesis given the current one? It has been shown empirically that people tend to update their hypotheses locally [1, 2], and I wonder if similar or different patterns hold for LLMs.

The authors mentioned the augmented model - the model that is equipped with the truth opponent strategies - still suffers in performance when the truth strategies are too complex. A growing body of work (eg, [3]) is playing with the idea of transforming natural language into programs (i.e. functions) to enhance validity of reasoning. I wonder if it is within the scope of this work - or perhaps future work - to improve LLMs’s ability to faithfully execute the math described in verbalized hypotheses.

Finally, while the modelling approach is well motivated, at a conceptual level, it seems to be more descriptive than purposeful - it is perhaps rational for people to trade-off between accuracy and complexity, where the cost of reverse-engineering a super complexity strategy can surpass the benefits of doing so. I wonder what the authors’ opinions are towards this trade-off in LLMs - should LLMs be free from such trade-off by design of its training regime (i.e. autoregressive next-word prediction does not necessarily trade off between length of a sentence and accuracy - sometimes longer context can even lead to higher accuracy)? It could be insightful in the discussion to highlight how this type of approach contributes to our understanding of either human behavior or internal dynamics of LLMs.

Minor: some citations are incorrectly placed. Eg, lacking parenthesis on line 655 for Binz & Schulz 2023.


[1] Bramley, Neil R., and Fei Xu. (2023). Active inductive inference in children and adults: A constructivist perspective. Cognition: 105471.

[2] Fränken, J. P., Theodoropoulos, N. C., & Bramley, N. R. (2022). Algorithms of adaptation in inductive inference. Cognitive Psychology, 137, 101506.

[3] Wong, L., Grand, G., Lew, A. K., Goodman, N. D., Mansinghka, V. K., Andreas, J., & Tenenbaum, J. B. (2023). From word models to world models: Translating from natural language to the probabilistic language of thought. arXiv preprint arXiv:2306.12672.

**Expertise:**

2

**Interest:**

3

---

> ### Author Rebuttal · Authors · 2025-04-15
>
> We appreciate your thoughtful comments and suggestions for potential extensions of the model. **In our revisions (blue text), we have added results evaluating model strategies, as well as discussion of how the model aligns with recent work on program synthesis and resource rationality.** See response to reviewers 1 and 2 for additional revisions.
>
> **Comparing human and model strategies:**
>
> The model’s natural language descriptions of opponent strategies represents an interesting avenue of evaluation. Because data from Brockbank & Vul (2024) includes post-experiment survey responses about the opponent’s strategy, HM’s hypotheses can be compared directly to people’s. To support this comparison, we identified representative hypotheses from human participants and HM that fell into four broad categories of accurate and inaccurate description. Humans and HM provide qualitatively similar responses across all four categories (see additional experiment 1 results and corresponding table in the supplement).
>
> **Mapping natural language to programs:**
>
> We were intrigued by the suggested use of program induction in the model. The current model augmentation results invite behavioral studies which could shed light on this possibility. Providing human participants with verbal descriptions of their opponent’s strategy may lead to improved performance against *all seven algorithmic opponents*. In this case, HM’s *failure* against the most complex opponent under similar conditions raises the possibility that a more structured representation of opponent strategies (e.g., a probabilistic program) is necessary to account for human behavior.
>
> **Model is “descriptive” rather than “purposeful”:**
>
> We share your interest in the ways HM might incorporate resource constraints faced by humans. Currently, HM does not seek to capture resource tradeoffs in this setting; our model augmentation and scaffolding results suggest that a core challenge for adapting to sequential opponent patterns in this context is *thinking of the right pattern* rather than merely *thinking harder*. However, comparing human behavior under various resource-limited or expanded conditions to language-based models with extended CoT or program synthesis represents an exciting opportunity for future work.
>
> **Minor (citations):**
>
> Thanks for catching the citation error on line 655. We have corrected it and done a careful parse of the remaining citations to make sure there are no other similar errors.

---

> > ### Comment · Reviewer_R5Wg · 2025-04-19
> >
> > The changes in the manuscript successfully address the issues raised.

---

### Official Review · Reviewer_qfDn · 2025-03-27
**This is a well-executed study of sequential reasoning, leveraging LLM-based models to capture human behavior in Rock, Paper, Scissors.**

**Soundness:** 3
**Clarity:** 2

**Comments:**

The manuscript investigates the use of LLM-based models to capture human behavior in Rock, Paper, Scissors. The authors introduce Hypothetical Minds, an LLM-based model designed to represent opponent strategies, and compare its performance to human players engaging with a set of algorithmic opponents. The key finding is that the model exhibits performance patterns closely aligned with those of human participants across different opponent types.

The study further explores ablated versions of the Hypothetical Minds model, revealing that the ability to generate hypotheses is a central bottleneck to performance. Notably, the authors demonstrate that incorporating pedagogically inspired interventions can lead to measurable improvements in the model’s effectiveness.

Strengths:
- The paper addresses a timely and relevant topic, likely to be of broad interest to the CCN community. It offers valuable insights that bridge cognitive science and AI, and suggests fruitful directions for future research.
- The claims are well-supported by the evidence and experiments presented.
- The manuscript is clearly written and well-structured, facilitating comprehension and reproducibility.

Weaknesses:
- No major weaknesses were identified.
- A few minor textual issues could be improved for clarity or polish

Suggestions / Questions:
- At line 186, when introducing the transition strategies, the distinction between positive and negative transitions may not yet be clear to the reader. It would be helpful to either briefly explain these terms or explicitly refer to Figure 1 for clarification.
- Could you clarify how critical the threshold of 0.7 is for validating a hypothesis? Was this threshold chosen empirically, or is there theoretical motivation behind it?
- Is there a formal derivation or proof that the Theory of Mind (ToM) module approximates Bayesian inference? If not, it would be helpful to better characterize the nature of this approximation.
- At line 320, the term action-based interaction is mentioned. A brief explanation or example would help readers unfamiliar with this concept.
- Can you speculate on the source of behavioral differences observed across different LLM base models? Are these primarily architectural, training-related, or emergent from differences in pretraining data?
- When increasing the model’s temperature, one would expect more diverse outputs, but potentially also a higher rate of lower-quality or hallucinatory responses. Did you explore or quantify this trade-off in your experiments?

I am reasonably confident in my assessment. While I am not an expert in cognitive science, I have a good background in AI and LLMs.

**Expertise:**

2

**Interest:**

3

---

> ### Author Rebuttal · Authors · 2025-04-15
>
> We thank you for the careful review and feedback. **The requested changes (blue text in manuscript) have significantly improved the manuscript’s clarity.** Below is a summary of these changes, along with requested clarification of LLM base model performance and temperature manipulation tradeoffs. See response to reviewers 1 and 3 for additional revisions.
>
> **Clarity changes:**
> * We have added text (line 178) clarifying positive and negative transitions with explicit reference to Fig 1.
> * We have clarified the motivation for the model’s value threshold of 0.7 (line 244). Briefly, this threshold corresponds to 85% prediction accuracy or roughly three correct guesses in a row. It was chosen to ensure that the model only considers fairly accurate hypotheses; in other domains, the threshold does not strongly impact results (Cross et al., 2024), though we do not explore that here.
> * Substantial additions to the Hypothetical Minds Architecture section (line 213) clarify the connection to Bayesian inference and planning models.
> * We agree that the phrase “action-based interaction” was unclear and have struck it from the relevant section (line 320). Updated text in that section clarifies how the ReAct model ablation chooses its move.
>
> **Different behavior across LLMs:**
>
> The behavioral differences observed across LLM base models likely stem from a combination of factors, including pretraining data, posttraining recipes, and architecture differences. The specific contributions of each of these is hard to investigate without direct access to the training data and procedures. We speculate that different models were exposed to varying amounts of *strategic reasoning*, *game theory*, and *pattern recognition* examples during pretraining and posttraining that affect their priors about strategic behavior in the current task.
>
> **Tradeoffs of increasing temperature:**
>
> When manipulating the temperature for GPT queries, we observed that higher temperatures increased the probability of *formatting errors* in the response. Our model handles this by re-prompting with the error when the response is incorrectly formatted. With GPT 4.0, responses were unusable with a temperature higher than 2.0 but the value of 1.0 in our temperature comparison balances this tradeoff: GPT 4.0 followed instructions and produced more diverse outputs, yet our results indicate that these were primarily lower quality hypotheses, leading to reduced performance.

---

### Official Review · Reviewer_P8x3 · 2025-03-27
**Why use a LLM for this?**

**Soundness:** 1
**Clarity:** 2

**Comments:**

In this manuscript the authors use LLM based agents to play rock paper scissors against simple agents with predefined patterns in their choices. In previous work humans had played against these agents, too and were able to exploit some of them but not others.  A very specific combination of inputs, LLM model and architecture around it performs somewhat similar to humans, while most others do not.

Unfortunately, this paper makes no sense to me. I don’t think there is much reason to apply LLM processing to a single task of this low complexity in the first place. Then the evaluation is really just based on average win rates against 7 opponents, i.e. on 7 numbers. Basing decisions about this many parameters on this little data is absurd. And even for these small data the authors’ model does not reproduce all patterns shown by humans and results vary a lot depending on detailed aspects of the model(s). Based on the results in the paper, I am actually more inclined to conclude that LLMs do various things depending on the exact prompts used and may not have anything to do with human behaviour. They certainly do not reproduce key features of human performance as claimed by the authors.

Detailed points:
- I don’t really see the advantage of using a large language model here. Maintaining a list of hypotheses about the opponent behaviour, evaluating those and planning out the choice are all processes for which models without any language exist that match humans quite well. This does not require a language model
- Using the large language models has many disadvantages here: As the authors note the results vary with the underlying LLM used, the exact formulations of their queries and other small variations making any successes extremely hard to interpret. Also tuning and evaluations is prohibitively expensive such that the models proposed cannot be optimised like simpler models can be. Also, it is an extreme computation overkill for this situation.
- If the models were able to capture some learning processes as displayed in Fig 2. or other more interesting patterns in the data this would also make things more interesting. Just matching the overall win rate is an extremely coarse pattern.
- For fitting 7 numbers of which 3 correspond to chance performance getting 1 categorically wrong is not good performance. Especially not for a complex models that combine multiple LLM processing steps.
- Many of the interesting observations the authors make about the LLM models are actually horrible aspects for a model of human behaviour: Giving scaffolds, i.e. adding slightly different prompts yields completely different results and some of the underlying language models fail to implement the interface right making unusable predictions.

P.S: I have nothing against using LLMs or other DNNs in cognitive models. Parsing complex situations or being task general may well require this, but for rock paper scissors as a single task, I really don’t see any reason to do this.

**Expertise:**

3

**Interest:**

2

---

> ### Author Rebuttal · Authors · 2025-04-15
>
> **We appreciate the opportunity to address your concerns and include additional analyses (blue text in manuscript) that strengthen the results.**
>
> **Model architecture:**
>
> We agree that the model should reflect the complexity of the underlying cognitive task. Even though the rules in RPS are simple, the space of opponent strategies is unbounded and can vary in complexity. Identifying which of the nearly infinite possible strategies an opponent is exhibiting based on their moves poses a computational challenge.
>
> One solution is to manually select the candidate opponent strategies and specify a distribution over them (comparable to our "Choose Hypothesis" model). This approach largely strips away the challenge of reasoning over an unbounded hypothesis space and fails to handle hypotheses that fall outside the specified distribution. Further, by encoding the possible strategies in ways that are specific to the RPS task, such models are unlikely to generalize to novel settings.
>
> Natural language is a useful and domain general way to parameterize the unbounded hypothesis space in this simple task. By combining the LLM with additional structures to approximate Bayesian inference over hypotheses, HM leverages the flexibility of natural language to create a domain general adaptive agent; the same model architecture can produce adaptive behavior across a range of cooperative, mixed-motive, and adversarial games (Cross et al., 2024), and math tutoring in parallel work. We have clarified this in the introduction.
>
> **Model evaluation:**
>
> You raise the important concern that win rate data is insufficient for evaluating the model. Two analyses in the revised manuscript address this limitation. First, we include learning trajectories, allowing for fine-grained assessment of HM’s ability to capture human behavior patterns. Second, we compare HM’s natural language hypotheses to those generated by participants in Brockbank & Vul (2024). When asked about their opponent’s most likely strategy after the experiment, participants provided qualitatively similar strategy descriptions to HM and exhibited similar error patterns.
>
> **Model scaffolding:**
>
> The concern raised about prompt sensitivity is that *arbitrary* differences between prompts produce different outcomes. The scaffolds provided in experiment 3 do not represent arbitrary differences from the original prompts. They were modeled after prior work in education; similar interventions might be performed with humans.

---

> > ### Comment · Reviewer_P8x3 · 2025-04-19
> >
> > I have read the authors response other reviews and updated manuscript. I have updated my relevance score up as the other reviewers clearly are interested in the paper a lot. Also the paper has improved, as the authors now describe a couple of things clearer and do compare to the learning trajectories at least a bit.
> >
> > I remain generally unconvinced by this paper though. In my view the created model does not explain the data well on an absolute scale and is vastly too complicated for this fairly simple task. I do not see any generalisable insights I gained from this model and for any potential interpretation an evaluation on rock paper scissors behaviour seems woefully inadequate to me.

---

> > > ### Author Response · Authors · 2025-04-22
> > >
> > > Thank you for your continued engagement with our paper and for raising your relevance score. In your response, you raise concerns that the model does not fit the data well on an absolute scale and is overly complicated for the task. In addition, you ask what generalizable insights can be gleaned from applying the model to the rock, paper, scissors task. **Below we address each of these concerns in the hopes you'll consider raising your soundness score.**
> > >
> > > First, our initial revisions sought to improve model evaluation in two ways: **quantitatively** by including learning trajectories and **qualitatively** by showing the similarity between human and model hypotheses (we note this latter result would be hard to achieve with a more traditional modeling approach that does not use a language model to sample the hypothesis space). If the addition of learning rates (which you explicitly mentioned in your response), as well as the comparison of natural language hypotheses does not improve your view of the paper’s soundness, perhaps you could clarify what additional analysis would address this as we seek to improve the paper?
> > >
> > > Second, in our previous response, we sought to address the concern that the model is overly complicated by observing that over 300 rounds of play against a strategic opponent, the model faces the nontrivial challenge of searching a practically unbounded hypothesis space in order to outwit the opponent. A model that samples natural language hypotheses is well-suited to this challenge and represents a reasonable hypothesis about what people do to solve the same problem. Do you have specific model architectures or approaches in mind that would be simpler while also addressing the central computational challenge of searching an unbounded hypothesis space?
> > >
> > > Finally, regarding generalizable insights, it is maybe more helpful to think about our work in terms of studying pattern recognition, rather than the children’s game of rock, paper, scissors. We note that this work is far from the first to use a relatively simple task domain in order to obtain more precise measures of human capacity and model fits (such simplifications are in fact desirable for these reasons). The RPS game is an ideal context in which to study human pattern recognition abilities in multi-agent settings where people need to adapt their policy to an opponent. The human results suggest that this cognitive ability is somewhat inflexible, that simpler patterns are learned rapidly, and that when faced with complex strategies, people often generate hypotheses far from the true strategy. Our model shows a similar (though not identical) response pattern and offers insights into why this might be the case, and how interventions might help a learner detect such patterns. In this way, our paper contributes to our understanding of the cognitive processes involved in such sequential pattern recognition and suggests that human alignment in pattern recognition will be an important benchmark for AI agents in the future.
> > >
> > > **Taken together, we hope you will reconsider in your soundness evaluation whether the paper “lacks critical evidence” or has “methodological flaws” or whether the points above might be considered the subject of more fine-grained disagreement about modeling approaches.**

---

### Meta-Review · Area_Chair_Jshv · 2025-05-04

**Ccn Recommendation:** Accept as Proceedings

**Metareview:**

While the concern about model complexity has merit, the authors have made a compelling case for their methodological choices. I agree that the use of natural language to parameterize the vast hypothesis space has merit and offers a domain-general approach with broad applicability in cognitive science. Additionally, the ablation and augmentation studies provide additional insights into hypothesis generation as a cognitive bottleneck in strategic reasoning. Given the detailed response from the authors, the additional analyses that address the core methodological concerns, and the strong positive assessment from two reviewers, I recommend acceptance to the Proceedings. The paper makes a substantive contribution to understanding human pattern recognition in strategic settings and demonstrates how LLM-based cognitive models can generate testable hypotheses about human cognition.

**Summary:**

This submission received three reviews with varying assessments. Two reviewers (qfDn and R5Wg) were very positive, rating the paper as clear and impactful. They appreciated the paper's methodology, analysis approach, and insights into decision-making processes. One reviewer (P8x3) was more critical, questioning the necessity of using LLMs for modeling a task as simple af rock-paper-scissors and expressing concerns about the evaluation metrics. The atuhors provided a detailed rebuttal addressing these concerns, clarifying their methodological choices, and adding some new analyses to the manuscript. Reviewer R5Wg confirmed that the changes successfully addressed their concerns. Reviewer P8x3 acknowledges some improvements but remained unconvinced about the model's explanatory value.

**Expertise:**

3